# HMGA Genes and Proteins in Development and Evolution

**DOI:** 10.3390/ijms21020654

**Published:** 2020-01-19

**Authors:** Robert Vignali, Silvia Marracci

**Affiliations:** Dipartimento di Biologia, Unità di Biologia Cellulare e dello Sviluppo, Università di Pisa, 56126 Pisa, Italy; silvia.marracci@unipi.it

**Keywords:** HMGA1, HMGA2, development, chromatin remodeling, AT hook, cell cycle, EMT, stemness, evolution, differentiation

## Abstract

HMGA (high mobility group A) (HMGA1 and HMGA2) are small non-histone proteins that can bind DNA and modify chromatin state, thus modulating the accessibility of regulatory factors to the DNA and contributing to the overall panorama of gene expression tuning. In general, they are abundantly expressed during embryogenesis, but are downregulated in the adult differentiated tissues. In the present review, we summarize some aspects of their role during development, also dealing with relevant studies that have shed light on their functioning in cell biology and with emerging possible involvement of HMGA1 and HMGA2 in evolutionary biology.

## 1. Introduction

HMGA proteins were initially isolated, together with other HMGs, by a biochemical purification procedure that allowed enrichment of small nuclear proteins from chromatin [1]. It was later discovered that HMGs were made of different classes of proteins; one of these was the HMGA (initially named HMGI) class of proteins [2]. At that time, studies on chromosome structure were unravelling the biochemical components of chromatin, and an α-protein that was able to bind to the primate α-satellite DNA was tentatively identified as corresponding to HMGI [3]. It was later shown that the α-protein had the DNA-binding abilities that were subsequently associated to a specific domain, the AT-hook domain typical of the HMGI/Y (now named HMGA1) and HMGI-C (now named HMGA2) proteins [4,5].

HMGA proteins are small, non-histone nuclear proteins that can bind DNA in the minor groove, and modify the chromatin conformational state and its accessibility by several regulatory factors, involved in the modulation of gene expression. In addition to their localization and associated nuclear functions, these proteins may also translocate into the mitochondria and bind to the mitochondrial DNA at the level of regulatory D loop sequences [6].

HMGA1 (in its splicing variants HMGA1a, HMGA1b, and HMGA1c, all encoded by the *HMGA1* gene) and HMGA2 (encoded by the *HMGA2* gene) are able to recognize the three-dimensional structure of specific regions of the double helix, usually (but not always) corresponding to AT-rich sequences [7,8]. Binding of HMGA proteins to DNA occurs in the minor groove [9] and is due to their AT-hook DNA-binding motif, whose consensus is the highly conserved amino acid sequence BBXRGRPBB (B=K or R residue; X=G or P residue) [10]. HMGA1 and HMGA2 usually have 3 or 4 such domains (Figure 1), with which they contact the DNA at their AT-rich target region [7,8,11]. More recently, RNA-HMGA1 interactions through the AT-hook domain of the protein have been identified, including the formation of RNA complexes with the viral transcript of HIV-1 [12]. Both HMGA proteins also possess an acidic tail (different in sequence between HMGA1 and HMGA2), that may serve to modulate their activity [13,14,15,16]. Apart from the AT-hooks and acidic tail, the two proteins do not show any special three-dimensional domain or any ordered structure; in fact, they are considered intrinsically disordered proteins, and it is generally assumed that this structural ‘freedom’ allows these proteins to bind DNA and modify its conformational state, as well as to interact with several other proteins [11,17]. Many of these latter proteins are transcription factors that HMGA1 and HMGA2 favor assembly into regulatory complexes, called enhanceosomes. For their ability to interact with many different molecular players, in a plethora of regulatory pathways, HMGA proteins have been regarded as a sort of “molecular glue”, or “hubs” for different nuclear functions, and have been connected to many aspects of gene regulation and of cell biology processes [17,18,19,20,21,22,23,24,25].

The activity of HMGA proteins is also tuned by post-translational modifications, such as phosphorylation, acetylation, and other modifications at specific residues (Figure 1). These modifications may be dependent on the intracellular or extracellular signals, so that the activity of HMGA proteins is tightly linked to internal and external influences [16,23,26,27].

Because of their biochemical function as chromatin architectural factors, HMGA proteins are involved in many aspects of development and differentiation, including proliferation, regulation and maintenance of stemness and potency, senescence, and chromatin state; they are also involved in regulating a key process in development, namely the epithelial-mesenchymal transition (EMT). Some of these abilities of HMGA may be recruited in the molecular dysregulation that takes place in tumor progression. In this review, we will mainly focus on the function of *HMGA* genes in physiological conditions, and often refer to data obtained in studies of cancer or other pathologies inasmuch they provide useful hints for understanding their developmental role. Other reviews have dealt more deeply with the involvement of HMGA in tumors and other pathologies [19,22,24,25,28,29,30,31].

## 2. Developmental Expression of *Hmga* Genes

### 2.1. Hmga1 Developmental Expression

*Hmga1* expression during mouse embryonic development was studied by in situ hybridization [32]. At E8.5 *Hmga1* transcripts are found in all embryonic tissues. Subsequently, its expression becomes more specifically localized and at E10.5 it is found in the central nervous system (CNS), including the brain and spinal cord, in the otic vesicle and the olfactory placodes; at this stage, strong expression is also detected in the somites, in the mandibular and maxillary arches, in the branchial arches and in several endodermal derivatives, including Rathke’s pouch, the developing stomach, liver, and pancreas. At E12.5-14.5 *Hmga1* expression remains high in the germinal zone of the brain (e.g., telencephalic vesicles), in sensorial structures as the olfactory epithelium and the retina, and in the dorsal root ganglia; other sites of high expression are the gut; the developing respiratory tract—including the bronchioli, the liver, thymus, thyroid, tongue, developing tooth primordia, the mesonephros, and the hair follicles. At E17.5, expression is maintained very high in the CNS, with special relevance for the telencephalic cortex and all the spinal cord; strong expression is also found in the spinal ganglia, in the retina and in the lens epithelium; HMGA1 transcripts are also detected in the epithelia of bronchioli, in the intestine, thymus, hair follicles, and in the seminiferous tubules of the testis.

A short description of *hmga1* developmental expression in zebrafish was published, reporting that *hmga1* transcripts are detected throughout embryogenesis and progressively become restricted to the most anterior areas of the embryo; only a low level of *hmga1* expression was detected in adult tissues [33].

### 2.2. Hmga2 Developmental Expression

*Hmga2* developmental expression has been studied by in situ hybridization and transgenic promoter analysis in the mouse [34,35] and by in situ hybridization in *Xenopus* early embryogenesis [36,37,38,39]. A very short description of *hmga2* expression in the zebrafish was published recently [40].

In the mouse, *Hmga2* transcripts were detected by in situ hybridization at E9.5 in almost all embryonic tissues, including the heart primordium and the developing spinal cord. In the following developmental stages, the mRNA distribution becomes more focused. At E12.5, *Hmga2* transcription is observed in tissues related to the nervous and sensory systems, such as the developing brain (ventricular zone of cerebral hemispheres and of the developing cerebellum), the nasal fossae, and the cochlea; in addition, *Hmga2* is also expressed in the neural crest derived mesenchyme of the developing maxilla and mandible and in mesodermal and endodermal derivatives, such as the forming ribs and vertebrae, the lung, stomach, gut, and kidney. Other sites of expression are the pancreas and the thymus [34]. At later stages (E14.5), expression is strongly maintained in the stomach and intestine, in the lung, in the kidney, in the cartilaginous tissue of the trachea, ribs and vertebrae, in the ventricular zone of the telencephalon and also in the diencephalic vesicle. Afterwards, *Hmga2* expression declines, though it is maintained at high levels at E17.5 in the crypts harboring intestinal stem cells, in the lung, and in the kidney [34]. Analysis of *Hmga2* expression with reporter constructs in the mouse confirmed most of this pattern of expression, but not all, possibly because of the absence of specific modulatory sequences in the reporter transgene [35].

On the whole, *Hmga2* transcripts are initially detected in most embryonic tissues and progressively become restricted to tissues that contain proliferative cells; *Hmga2* expression is usually absent in adult tissues, except for the testis (where it seems required for meiosis of the spermatocytes), the lung, and the kidney [41,42,43]. Specific details of the expression of this gene or protein in the mouse have also been reported in studies on their roles in the developing neural tissue and other tissues [44,45,46,47,48,49,50,51]; we will deal with some of these in the following sections.

In the frog, *Xenopus laevis*, localized *hmga2* mRNAs are first detected as maternal transcripts that are relocated to the animal pole during the initial cleavage stages [37]; after the midblastula transition, *hmga2* mRNA is detected in the neural plate, in the pre-placodal region and in the presumptive neural crest of the early neurula; at later neurula stages, *hmga2* mRNA can be detected in the closing neural tube and migrating neural crest cells (NCCs), as well as in the precardiac region; during the subsequent tailbud stages, *hmga2* mRNA persists in the neural tube, is found in the neural crest derived pharyngeal arches, in the otic vesicle, in the notochord, in the heart forming region, in the intermediate mesoderm and, at a lower level, in the developing eye [36,38,39].

A recent report has analyzed very succinctly the pattern of expression of *hmga2* in zebrafish, concluding that the mRNA distribution of *hmga2* transcripts parallels that of *snai1*, reported to be expressed in the primitive veins and in the head region at 20 hours post fertilization (hpf), in the developing posterior cardinal veins at 24 and 30 hpf, and in the heart at 48 and 72 hpf [40]. *snai1* is expressed in the developing pharyngeal arches [52], so it looks as if zebrafish *hmga2* pattern of expression is quite similar to that of *Xenopus*.

A graphic synopsis of *Hmga1* and *Hmga2* developmental expression in a generalized vertebrate embryo is presented in Figure 2.

## 3. Developmental Roles of *Hmga* Genes and Proteins

### 3.1. HMGA Dysregulation and Its Impact on Body Size

A first idea of *Hmga* function in development came out from studies performed on knockout mice and on mice overexpressing truncated or wild-type forms of the *Hmga* genes.

The first *Hmga* gene to be inactivated by homologous recombination in the mouse was *Hmga2* [53]. *Hmga^−^*^/*−*^ mice display the *pygmy* phenotype, with reduced overall size, that had been described earlier [54]; at 8–10 weeks, their weight is about 40% of the normal value, but weight reduction is already detectable at E15.5 [53,54,55,56]. Interestingly, *Hmga2* loss-of-function has recently been accomplished in the pig, and substantially confirms the great reduction in body size [57]. In addition, *Hmga2* depletion caused sterility in both mouse and pigs. In particular, *Hmga2*^−/^^−^ pigs show cryptorchidism, and hence the sterility is possibly due to absence of spermatogenetic cells caused by undescended testes [57]; whereas, transgenic *Hmga2*^−/^^−^ mice have normally descended testes but no mature sperms because of impaired spermatogenesis [43].

These effects of *Hmga2* inactivation on body size may depend on *Hmga2* influence on proliferation and differentiation of diverse tissues (see Section 3.4). The impact of *Hmga2* loss-of-function on body size is further exacerbated by the combined loss-of-function with the *Hmga1* gene [58]. While at 1 year of age, some 27% reduction in body weight is observed in *Hmga1* knockout mice (compared with 55% reduction in *Hmga2* KO), the double KO results in the so called “superpygmy phenotype”, with an impressive 75% reduction in weight. The reduced number of double mutant embryos recovered in these experiments, compared to expected, indicates some lethality; in addition, most born double knockout mice die within 1 year of age [58]. Analysis of mouse embryonic fibroblasts (MEFs) from these embryos showed that there is an increase in the levels of p16, p19 and p21, and an enhanced susceptibility to senescence. On the contrary, there was a strong reduction in the levels of active acetylated E2F1, that became not detectable, and of Cyclin A and Cyclin E, which are known targets of E2F1 and are involved in cell cycle progression [59,60]. These effects, already detectable in the single KO, are much stronger in the double KO mice and may explain the observed phenotype [58].

The consequences of these loss-of-function on the size of specific tissues/organs may not be proportional to overall size reduction: fat and skeletal muscle tissue are in fact disproportionately reduced, compared to other tissues, that may be diminished on the same scale of overall reduction, or even increased [56].

The effects on size may be related to the activity of HMGA proteins in sustaining proliferation and/or differentiation of specific tissues, in particular of fat and skeletal muscle (see Figure 3A and following text). *Hmga2*^−/−^ mice show a reduction in fat tissue [55], due to lower proliferation of pre-adipocyte cell precursors [61]; significantly, *Hmga2* expression is reactivated in adult fat tissue of wild type mice upon high fat diet, and in leptin-deficient genetic models of obesity [61]. The involvement of *Hmga2* in fat tissue differentiation is also stressed by the observation that *Hmga2* deficiency renders the mouse resistant to diet-induced obesity and mitigates the effect of leptin deficiency [61]. Consistent with these observations, overexpression of truncated or wild type forms of *Hmga2* leads to excess of fat tissue and development of lipomas [62,63]. In vitro studies further support the role of *Hmga2* in adipogenesis. *HMGA2* is readily induced by FGF and PDGFBB in 3T3-L1 cells, an in vitro model for adipogenic differentiation [64]. HMGA2 participates in activating a program of adipogenesis by recruiting C/EBPβ onto the regulatory region of *PPARγ*, a crucial gene for adipogenesis [65]. HMGA2 and STAT3 cooperate in maintaining a proliferative state in 3T3-L1 cells [66], STAT3 is also involved in the regulation of transcription of *C*/*EBPβ* and *PPARγ* [67,68,69]. In human pre-adipocytes, *SREBP1* and *SREBP2* (encoding the two transcription factors known as sterol regulatory-element binding protein 1 and 2) are key genes for regulating fatty acid metabolism and in vitro adipocyte differentiation; their mRNAs are co-transcribed together with mir-33b and miR-33a, respectively [70]. Interestingly, these two miRNAs target *HMGA2*, *CDK6* mRNAs and *Cyclin D1*, but it is the downregulation of *HMGA2* that appears particularly important to impair proliferation of pre-adipocytes and consequent fat tissue formation. Similar effects to those of miR-33b overexpression, though somehow weaker, were obtained upon siRNA *HMGA2* interference, suggesting that a great deal of miR-33b effects are mediated by *HMGA2* [70,71]. In addition, HMGA2 levels are also subject to the translational control of let-7: following the clonal expansion of pre-adipocytes, HMGA2 is downregulated by increasing let-7 levels to allow transition from the proliferating phase to the differentiation process [72].

*Hmga1* overexpression in adipose tissue, induced by the *Ap2* promoter in transgenic mice, caused a marked reduction in white and brown adipose tissues associated with downregulation of the mature adipocyte markers such as *Pparγ C*/*ebpβ*, *C*/*ebpα*, *Fabp4*, *Adipoq*, and upregulation of pre-adipocyte markers like *Pref1* [73].

Therefore, it seems that, similarly to muscle (see below), HMGA proteins sustain the early proliferative phase of precursor expansion and participate in initiating the early expression of differentiating genes, such as *Pparγ*, but then need to be downregulated in the phase of terminal differentiation.

*Hmga2* ability to impact on size and weight is also demonstrated in muscle tissue. Together with genes involved in cell cycle regulation, *Hmga2* emerged in a group of genes that constitute a ‘myoblast signature’: proliferation of myoblasts and muscle development are impaired in *Hmga2* deficient mice [56]. Accordingly, ex vivo studies show that muscle satellite cells induced to become myoblasts strongly upregulate *Hmga2* expression, that is rapidly downregulated in their subsequent differentiation to myotubes. Consistent with this and with a possible crucial role of *Hmga2* in skeletal muscle development, *Hmga2* is expressed in proliferating myoblasts in vivo. Significantly, *Hmga2^−^*^/*−*^ mice show a severe reduction in skeletal muscle growth in the postnatal period, due to a great diminution of proliferating myoblasts, as assessed by the reduced number of PAX7 expressing muscle progenitors and of Ki67+ cells, roughly half than in normal mice. Besides the effects on proliferation, when satellite cells were isolated from muscle fibers of *Hmga2^−^*^/*−*^ mice and assayed in culture, they showed a severe restriction of their self-renewal potential [56]. A substantial part of these effects is due to downregulation of *Igf2bp2* (*Imp2*), a member of the insulin-like growth factor binding proteins and a described direct target of HMGA2, not only in muscle precursors, but also in other cellular contexts [74,75,76]. In fact, interfering with *Igf2b2* function in myoblasts leads to similar effects as *Hmga2* depletion; on the other hand, the deficiency in proliferation observed in *Hmga2*^−/−^ myoblasts is rescued upon overexpression of *Igf2bp2*. Consistent with this, HMGA2, together with p65, directly binds to regulatory sequences in the first intron of *Igf2bp2* to enhance its transcription [56,75]. IGF2BP2 is an RNA binding protein that positively regulates the translation of mRNAs that promote proliferation, like *c-Myc*, *Sp1*, *Igf1r* and others [56]. Significantly, C-MYC, SP1 and IGF1R proteins were reduced both in *Hmga2^−/−^* myoblasts and in *Igf2bp2* depleted myoblasts, though the levels of the corresponding mRNAs were unaffected (see Figure 3A for a resuming scheme). It also turned out that following proliferation and early myogenesis HMGA2 levels must be downregulated to allow differentiation; similar requirement holds true also for C-MYC [56,77,78,79,80].

HMGA1 is also involved in muscle development, where it prevents differentiation by repressing myogenic gene expression. Murine C2C12 cells can be committed to initiate muscle differentiation upon growth factor retrieval. After induction of myogenesis, both *Hmga1* mRNA and protein, normally expressed in these cells, are immediately downregulated. When *Hmga1a* is overexpressed in C2A1 cells (a C2C12 line that stably expresses *Hmga1a*), they fail to initiate the myogenic program: they express high levels of the repressor of myogenic differentiation *Msx1,* while *MyoD* and *Myogenin,* that are positive regulators of the myogenic program, are downregulated. Besides, the genes for the insulin-like growth factors *Igf1* and *Igf2* and the insulin growth factor binding proteins *Igfbp2* and *Igfbp3*, required for proper myogenesis, were also downregulated in these cells. When *HMGA1a* was downregulated in C2A1 cells by siRNA interference, appropriate expression of these genes was rescued, and the myogenic program was resumed [81]. This suggests that, similar to HMGA2, HMGA1 downregulation is essential for the progression of the myogenic program. HMGA1 induces in myoblasts the expression of ID3, one of the major repressors of cell differentiation, that presumably maintains cells in a myoblast stage [82]. Interestingly, in mouse differentiating cells, miR-195 and miR497 repress *Hmga1* expression by targeting its 3′ UTR, and in turn ID3 is downregulated. Overexpression of miR-195/497 or *Hmga1* silencing in C2C12 cells promotes myotube formation and myogenic differentiation. This suggests a novel regulatory axis for myogenesis in which miR-195/497 promote myogenic differentiation by repressing the HMGA1-ID3 pathway [83] (Figure 3A).

Furthermore, the ε isoform of PKC serine-threonine kinase (PKCε) also downregulates *Hmga1* expression, consequently leading to an increased expression of myogenic differentiation genes in mouse. In the nucleus, PKCε blocks *Hmga1* expression and triggers two key genes for muscle differentiation, *Myogenin* and *Mrf4*, indicating the PKCε-HMGA1 axis as a regulator of skeletal muscle differentiation [84] (Figure 3A).

In addition to these molecular evidences, a series of recent studies indirectly confirms that both *Hmga2* and *Hmga1* are involved in size/weight/height control in several different species. A rather well characterized example of *Hmga2* mutant is that of the dwarf phenotype in rabbits [85,86]. The *dwarf* mutation is a 12.1 kb deletion that removes the promoter and the three first exons of the rabbit *Hmga2* gene; as the name implies, the mutation leads to reduction in size, shorter limbs, smaller ears and reduced weight; in addition, the mutation is also accompanied by craniofacial dysmorphologies [87]. However, differently from the *hmga2^−/−^ pygmy* phenotype in mice, the dwarf mutant rabbit is heterozygous for the deletion, implying that the mutation is semi-dominant. Rabbits that are homozygous for the *dwarf* deletion die shortly after birth and have an extremely reduced body size and abnormal head shape. RNA-seq analysis shows that *Igf2bp2* is downregulated in the homozygous *dwarf/dwarf* condition, though not in the heterozygous *Dwarf/dwarf*, with respect to the *Dwarf/Dwarf* condition, confirming the functional connection between *Hmga2* and *Igf2bp2* [87].

In chicks, dwarfism has been associated with mutations in a genomic region containing *Hmga2* and *IGF1* [88,89].

Several genome-wide association studies (GWAS) have outlined HMGA2 as an important factor in determining dimensions in humans, horses and dogs [90,91,92,93,94,95,96,97,98,99,100,101]. Furthermore, *Hmga2* mutations have been involved in Silver–Russell syndrome (SRS), or SRS-like syndromes [102,103,104], or ear dimensions in pigs [105]. A search for candidate genes involved in SRS, a rare form of fetal growth retardation usually due to downregulation or malfunctioning of *IGF2*, has pointed to *HMGA2*, *PLAG1*, and *IGF2* as genes causally involved, and functional experiments show that HMGA2 promotes *IGF2* expression in PLAG1-dependent as well as PLAG1-independent ways; this underlies the significance of this pathway (see also Section 3.2) in the regulation of growth during the fetal and postnatal periods [102] (Figure 3A).

Similarly, for *Hmga1*, GWAS have uncovered new genetic variants associated with phenotypic external traits in different races of pigs. *Hmga1* was indicated as a relevant candidate for bone limb length, for fatness and for large or small body size in different brands of Chinese pigs [106,107,108,109,110]. HMGA1 may act as a modulator of IGF1 (insulin-like growth factor 1) pathway activity, based on the fact that it is able to influence the expression of IGFBPs [111,112]. Interestingly, GWAS performed in human subjects reported that *HMGA1* and *IGFBP4* may be ideal candidates of specific anthropometric traits such as adult height and body mass index [113,114,115].

### 3.2. Hmga1 and Hmga2 in the Development of the CNS

As already remarked, *Hmga2* is expressed in the early development of the mouse CNS; localized mRNAs are first detected at E9.5, but then its expression is already declining at E14.5 and the following stages [34,44]. In the forebrain, *Hmga2* (and *Hmga1*) transcripts are detected in the ventricular zone (VZ) of the telencephalic vesicle, both in the pallial and subpallial areas [50]. *Hmga2* role has been addressed in more details in the pallial derivatives, especially the neocortex.

Brains of *Hmga2^−/−^* mice are smaller compared to controls. When neural stem cells (NSCs) were dissociated from the early forebrain (E11.5 and E14.5) or late forebrain (P0 and P49–56) of *Hmga2^−/−^* mice and compared to wild type stem cells from equivalent stages for their ability to form primary multipotent neurospheres, it was observed that those from early (embryonic) stages were as efficient as wild type cells, while those from later (post-natal) stages showed a reduced efficiency compared to control cells. In addition, secondary neurospheres generated from both early or late forebrain *Hmga2^−/−^* stem cells were smaller in size and had lower self-renewal ability compared to neurospheres generated from their respective wild type control stem cells. These data suggest that *Hmga2* function is required to maintain the self-renewing capacities of forebrain stem cells, though not for their initial establishment at early stages. Further observations suggest that the impairment of self-renewal may be due to reduced proliferation, while apparently there is no effect on the differentiative potency of NSCs, as no distinctions were found in their ability for multilineage differentiation [44].

The ability of *Hmga2* to promote and maintain NSC renewal is due to its negative regulation of *p16^lnk4a^* and *p19^Arf^*: in fact, these genes are upregulated in *Hmga2* defective neural stem cells, and single or combined inactivation of *p16^lnk4a^* and *p19^Arf^* in *Hmga2^−/−^* neurospheres partially restores the normal capacity of neural stem cells, both in the CNS and in the peripheral nervous system (PNS) [44]. Even more significantly, in vivo, *Hmga2^−/−^* mice show reduction of proliferation in the sub-ventricular zone, reduced number of some peripheral neurons and reduced brain size, but all these effects are partially rescued by the combined inactivation of *p16^lnk4a^* and *p19^Arf^*; however, the overall body size reduction caused by the *Hmga2^−/−^* genotype was not rescued, suggesting that the *p16^lnk4a^* and *p19^Arf^* downregulation fulfilled by *Hmga2* in the CNS and PNS may not apply to other tissues. This could be consistent with the observation that p16 and p19 are unaffected in *Hmga2^−/−^* mutant muscle cells [56].

The negative regulation of *p16^lnk4a^* and *p19^Arf^* by *Hmga2* appears to be indirect and may be mediated through JUNB, a described activator of *p16^lnk4a^* and *p19^Arf^* [116,117] (Figure 3B); in fact, HMGA2 binds to the *JunB* promoter and negatively regulates its transcription [44]. Finally, it was found that the requirement of *Hmga2* function for self-renewal abilities of NSCs is overcome at old-adult stages, when its role in repressing *p16^lnk4a^* and *p19^Arf^* is taken over by *Bmi1* [44].

One important aspect of *Hmga2* action concerns its regulation, both at the transcriptional and at the post-transcriptional level. This control appears of pivotal significance in the brain, given recent evidence that points at *Hmga2* as involved in a temporal program that on one side dictates the timing of neurogenesis versus gliogenesis, and, on the other side, sets the time for the orderly generation of the different layers of the developing mammalian neocortex [118,119].

Although evidence for a direct regulation seems missing, the HES5 transcriptional repressor is able to negatively regulate *Hmga2* (as well as *Hmga1*) transcription [50]. Mice overexpressing *Hes5* show reduced body size, reduced brain size, lower proliferation of neural progenitors and reduced thickness of the cortex. These effects recall those of the *Hmga2^−/−^* mice [44]; however, concurrent downregulation of *Hmga1* may also contribute. When *Hes5* is overexpressed, generation of later and more superficial neurons, as well as gliogenesis, is anticipated; on the contrary, in *Hes5^−/−^* mice, they are delayed. Correspondingly, *Hmga2* (and *Hmga1*) transcription is downregulated in the *Hes5* overexpressing neocortex, while it is upregulated in the *Hes5* knockouts [50] (Figure 3B).

The consequences of *Hes5* overexpression or loss-of-function on the timing of neurogenesis vs. gliogenesis are consistent with other data showing that *Hmga1* and especially *Hmga2* are involved in regulating the temporal aspects of cortical development. In particular, interfering with both *Hmga2* and *Hmga1* in early neural precursor cells (NPCs) leads to impaired neurogenesis and increased astrogenesis, implying a precocious restriction of the differentiative potential of early NPCs [120]. On the other hand, overexpression of *Hmga1* or *Hmga2* gene in early NPCs prolongs their competence to generate neurons rather than astrocytes (that is neurogenesis vs. gliogenesis), while their overexpression in late NPCs is even able to revert them to an early competence stage, leading to an increased production of neuron-containing clones vs. astrocyte-containing ones [120].

Besides these evidences obtained on in vitro cultured NPCs, in utero electroporation of siRNA vectors for both *Hmga1* and *Hmga2* in early stage (E15.5) NPCs reduces the percentage of differentiating neurons, and leads to fewer, late born, CUX1 positive neurons in the population of transfected cells; conversely, in vivo overexpression of both *Hmga1* and *Hmga2* increased the number of differentiating neurons, extending the neurogenetic time-window and leading to the generation of more late-born CUX1 positive neurons than in control brains. Finally, overexpression of both *Hmga* genes is even able to promote neurogenesis in postnatal P1 NPCs, that would normally produce astrocytes, thus reverting their competence to an earlier neurogenic potential [120].

The role of HMGA proteins in cortical development is carried out through their chromatin remodeling activity. Overexpression of *HMGA1* and *HMGA2*, or their depletion, makes chromatin of in vitro NPCs more prone or more resistant to digestion by meganuclease (MNase), respectively. In vivo electroporation of the two genes at E15.5 induces an increase of TBR2 positive cells at P6 or P8, compared to controls; however, when *H1cc* (encoding a mutant histone H1 that leads to a tighter chromatin compaction compared to wild type H1), is co-electroporated with *HMGA*, sensitivity to MNase diminishes and a lower number of TBR2 positive cells are generated, compared to NPCs expressing only HMGA [120].

What are the genes that HMGA chromatin remodeling activity helps to recruit into the genetic program leading to cortical neurogenesis? Two such genes have been identified in *Igf2bp2* and *Plag1* (Figure 3B).

*Igf2bp2* is a target of HMGA2 in different cell types [56,74,75]; it is an mRNA binding protein also involved in stem cell maintenance in glioblastoma [121], a neural tumor in which *HMGA2* is reactivated [122,123,124,125]. Similar to *Hmga2*, *Igf2bp2* is initially expressed at high levels in cortical NPCs in the early neurogenic phase, and then downregulated in late NPCs in the gliogenic phase. *Igf2bp2* expression is increased by *Hmga2* overexpression in NPCs, while depletion of *Hmga2* reduces *Igf2bp2* levels of expression [76,126]. *Igf2bp2* overexpression in late NPCs (E17.5 + 3DIV) prevents their differentiation into astrocytes, instead favoring a neurogenic potential [76]; on the other hand, depletion of *Igf2bp2* in early NPCs (E11.5 + 3DIV) favors gliogenesis and impairs neurogenesis, without any variation in proliferation or in cell death [76]. *Igf2bp2* is therefore able to mediate, at least in part, the activities of *Hmga2* in NPCs. In glioblastoma cells, it has also been shown that *Igf2bp2* can protect let-7 target molecules from being silenced by this miRNA, suggesting a new mechanism of regulation for maintaining the NSC state [121].

PLAG1 is a zinc-finger transcription factor, expressed in various tissues including the telencephalic VZ of the mouse, where it is required for normal development of the neocortex [127,128]. Again, similar to *Hmga2*, the expression of *Plag1* in NPCs declines as they proceed along development [126]; moreover, HMGA2 binds directly to the promoter of *Plag1*, whose expression is upregulated or downregulated, respectively, by gain- or loss-of-function of *Hmga2* in NPCs [126]. In vitro experiments on early stage (E11.5 + 3DIV) NPCs overexpressing Plag1 show that these cells produce a larger proportion of neurons compared to controls, while similar overexpression in a later astrogenetic phase (E11.5 + 9DIV) causes cells to rescue their neurogenetic potential and produce larger proportion of neurons and fewer astrocytes. On the contrary, depletion of *Plag1* in NPCs at E11.5 + 3DIV increases astrocytes. Consistent with these results, in vivo experiments confirm the observations made on cultured NPCs: in utero electroporation experiments show that *Plag1* overexpression promotes neuronal differentiation in NPCs, while interference on *Plag1* function does the opposite [126]. By transcriptomic analysis, a set of 46 genes, most of which involved in neuronal development, was downregulated in vivo by interfering with *Plag1* function, while 1 gene was found upregulated; for many of them, a PLAG1 binding motif was found in the surroundings of their transcription start site [126].

In conclusion, *Hmga2* may coordinately regulate the neurogenic potential of NPCs both by stimulating *Igf2bp2* activity and by facilitating *Plag1* expression and in turn of its many targets (Figure 3B).

The activity of *HMGA2* in cortical neurogenesis is matched with the highly precise timing of neuronal generation [119]. Micro RNAs (miRNAs) are often involved in developmental processes where timing of gene expression needs to be tightly linked to cell fate decisions and/or sequential tissue layering, as in the developing neocortex and retina [118,119,129,130].

The three miRNAs mir-9, mir-128, and let-7 are expressed according to dynamic temporal gradients that regulate layer formation in the developing neocortex. Expression of miR-9 and miR-128 was found to decrease, while that of let-7 was found to increase, during cortical development [51]. *Hmga2* mRNA has let-7 target sites in its 3′UTR and therefore this miRNA prevents production of the HMGA2 protein in several cell types [28,131,132,133]. The temporal dynamic of its expression suggests that it may work as a negative regulator of *Hmga2* expression in the cortex. In particular, as monitored by in situ hybridization, let-7b and let-7c expression was undetectable at E11.5, but was progressively upregulated from E12.5 onwards, during neurogenesis; by microarray analysis, this was confirmed and extended to other members of the same family (let-7a and let-7g). At P1, let-7 expression forms a gradient in upper layers II-IV, being more abundant in layer II than in layer IV. Consistent with this, let-7 overexpression by in utero electroporation of developing cortices at E12.5, E13.5, or E14.5 increased the number of transfected cells that were found in upper layers IV-II at P3, and diminished the number of cells in deeper layers V-VI, compared to control cortices. On the other hand, depletion of let-7 miRNA led to an enrichment of transfected cells in the deeper layers. In both let-7 overexpression or depletion, these changes of layers in transfected cells were accompanied by a change in gene expression. It seems that the miRNAs do not have an instructive effect as to the cell fate per se, but only serve to address the cells to the appropriate layer that is being made at that time. In fact, let-7 overexpressing, or let-7 depleted cells, shift their laminar position towards an upper or deeper layer, respectively, but their final position and cell fate depends on the day when the NPCs where electroporated (that is, on their neurogenic state) [51].

These results are consistent with the idea that let-7 and *Hmga2* are part of a temporally regulated program that controls the timing of cortical layering, as well as the neurogenic potential of NPCs, by post-transcriptional downregulation of *Hmga2* driven by let-7 (Figure 3B).

Within this program, other important players appear to be the two mRNA binding proteins LIN28 and IGF2BP1 (insulin growth factor2 mRNA binding protein 1, another member of the IGFBP superfamily). Both these mRNA binding proteins favor HMGA2 expression. LIN28 is able to inhibit let-7 miRNA biogenesis in different contexts (see below), therefore favoring *Hmga2* mRNA translation into protein. Although let-7 mediated inhibition of *Hmga2* has been demonstrated, LIN28 may also promote HMGA2 expression in a let-7-independent way [49]. The effects of LIN28 abrogation were studied in *Lin28a^−/−^* single knockout mice and *Lin28a*^−/−^/*Lin28b*^+/−^ compound mice (double knockouts are early lethal). *Lin28a^−/−^* mice show reduced body dimensions and reduced brain size; in the cortex, thickness is reduced, due to reduced proliferation and reduced numbers of NPCs; these defects are intensified in *Lin28a*^−/−^/*Lin28b*^+^^/−^ genotypes. On the contrary, overexpression of *Lin28a* under the control of the *Nestin* promoter produced enlarged brains, an increased thickness of the neocortex, and an increase of proliferation and of NPCs. However, let-7 levels were found unvaried in the brains of the compound *Lin28a*^−/−^/*Lin28b*^+^^/−^ mice. Further experiments showed that LIN28a is able to interact with IGF2BP1, and that *Hmga2* mRNA and a handful of other mRNAs (involved in the IGF-mTOR pathway) are bound to LIN28a in RNA immunoprecipitation experiments. One model has been proposed according to which LIN28a and IGF2BP1 bind to *Hmga2* mRNA to facilitate its translation, thus promoting the proliferation and self-renewal properties of early progenitors associated to HMGA2 activity [49].

*Hmga2* is also involved in regulating self-renewing abilities of retinal progenitor cells (RPCs) [46]. In experiments performed in rats and aimed to test possible molecular pathways necessary to regulate RPC differentiative potentials, endothelial cells were found to release signals that allow late (E18) RPC-derived neurospheres to be cultured at low density conditions. In a transcriptomic analysis, *Hmga2* was found among the genes enriched in RPC-derived neurospheres grown in endothelial conditioned medium. The response to the conditioned medium, both in terms of numbers of generated neurospheres and in terms of *Hmga2* levels of expression, was abolished by a cocktail of inhibitors, or single inhibitors, of those signaling pathways that the transcriptomic analysis had found to be upregulated by the treatment. In the developing retina, *Hmga2* is expressed according to a similar declining temporal dynamic as that observed in the brain: high expression at E14, low expression at E18, very low expression at P3, no expression in the adult. When neurospheres derived from E18 RPCs were grown in conditioned medium, *Hmga2* was upregulated and let-7 downregulated; administration of the cocktail of signaling pathway inhibitors led to the opposite effect. Significantly, a series of experiments in vitro (neurosphere assays), in vivo (intravitreal retroviral infections) and ex vivo (retinal explants) also showed that *Hmga2* gain-of-function was able to increase self-renewal capability of late RPCs in the absence of conditioned medium; on the contrary, loss-of-function of *Hmga2* in similar in vitro and ex vivo experiments abrogated the self-renewal of late RPCs promoted by the conditioned medium, as well as their in vivo self-renewal. In all these experiments, analysis of the expression of JunB and p19^arf^, that we mentioned earlier as part of the same regulatory axis in cortical NPCs, was consistent with the results obtained in CNS cell precursors [44].

Further extending this work, it was also showed that the changing equilibrium of the let-7-HMGA2 axis is also employed in the retina for regulating the switch from early neurogenesis to late neurogenesis and gliogenesis [134]. The expression of the let-7 family members was monitored and it was observed that their expression increases steadily during development until the adult stage. This temporal pattern is basically opposite to HMGA2 expression. When let-7 function was antagonized by shRNA interference, the differentiation of late cell types (rods, bipolars, and Muller glia cells) was impaired; on the other hand, the gain-of-function approach, produced exactly opposite results, showing an increase of late born cell types in the cultures. The effects of let-7 overexpression are counteracted by a let-7 resistant form of *Hmga2* [134].

The emerging and general model is consistent with what seen above for the neocortical NPCs: LIN28 and HMGA2 favor NPCs and RPCs self-renewal; some of the effects of LIN28 may be mediated by downregulating let-7 and therefore indirectly upregulating HMGA2; some may be let-7 independent and, as outlined above, may imply IGF2BP1 stabilizing *Hmga2* mRNA and favoring its translation. When let-7 raises up, HMGA2 protein production, as well as those of LIN28 and IGF2BP1, are downregulated and cells are addressed to cell cycle exit and differentiation, rather than to self-renewal, through releasing the repression of *Jun B* and of *p16^lnk4a^*/*p19^arf^*(Figure 3B).

The role of *Hmga1* in the developing retina has not been completely clarified. A recent report described *Hmga1* expression in the mouse retina by a combination of transgene reporter analysis, in situ hybridization and immunostaining. These experiments lead to conclude that *Hmga1* expression in the retina is especially strong at E16, both in proliferating cells and in committed neurons, while later, at P2, is essentially found in the central and inner retina. *Hmga1* expression was then detected at 14 weeks of age in all main neuronal cell types of the inner retina, with the exception of photoreceptors, where expression is said to be rarely detectable [135]. These data are somehow at variance with those obtained earlier by other authors, showing HMGA1 protein expression in the mouse photoreceptors and cooperative action with the CRX transcription factor to elicit transcription of the rhodopsin gene [136].

### 3.3. Hmga Genes in Xenopus Development: Focus on Neural Crest and Heart

The developmental expression and role of *hmga*-related genes were also studied in the frog, *Xenopus laevis*. *hmga2* is the only canonical member of the family present in *Xenopus*: both *Xenopus tropicalis*, that is diploid, and *Xenopus laevis*, that is pseudotetraploid [137,138] do not harbor a true *hmga1*. However, three divergent *hmga*-related multi-at-hook cDNAs (*hmg-at-hook1*, *2* and *3*) were described in *Xenopus laevis*. *hmg-at-hook1*, *2* and *3 (hmg-at-h1*, *hmg-at-h2* and *hmg-at-h3)* code for proteins containing 8, 8, or 6 AT-hook domains, respectively; differently from *HMGA*, none of these cDNAs encode a terminal acidic tail. Consistent with these different features, Hmg-at-h1 protein showed quite distinct biochemical properties compared to HMGA1 and HMGA2 [139]. *Hmg-at-h1*, *2* and *3* cDNAs are all potentially encoded in the single locus existing in *Xenopus tropicalis*, as well as in the duplicated loci of *Xenopus laevis* [139]. The expression of these cDNAs was analyzed during embryogenesis; basically, *hmg-at-h2* is undetectable, *hmg-at-h1* is more abundant as a maternal transcript and then levels off, *hmg-at-h3* is expressed at low levels initially and then becomes upregulated during the neurula stages. Localized transcripts are detectable at early tailbud stage in the developing CNS (including the eye) and in the neural crest and its derivatives [139]. The functional role of these proteins was studied by morpholino (MO) antisense injections targeting all the three potential mRNAs, for single or combined ablation. Effects were very mild or absent when MOs for each transcript were injected separately; when MOs against the two most expressed mRNAs, *hmg-at-h1* and *hmg-at-h3*, where combined together, a small, but detectable, disturbance was observed on the cranio-skeletal derivatives of the branchial arches and on the dimensions of the eye vesicle, and was accompanied by reduced expression of relevant neural crest markers [139].

The phenotypic effects on craniofacial derivatives of NCCs were much stronger when *hmga2* function was interfered [39]. As already remarked, localized *hmga2* transcripts are first detected in the developing neural plate, prospective neural crest and prospective pre-placodal ectoderm, and subsequently found in the brain, spinal cord, NCCs, otic vesicle. NCCs are a special population of multipotent cells, that undergo an EMT and then migrate away from their initial position at the neuroectodermal–epidermal boundary to reach different districts in the developing embryo, where they will differentiate into a variety of neural and non-neural tissues [140]; cranial NCCs, for example, will give rise to craniofacial skeletal derivatives and contribute to the different facial appearance of each vertebrate species [141]. Anti-*hmga2* MOs (MO-*hmga2*) injections targeting the future anterior neural plate and neural crest forming region, lead to severe disruption of the craniofacial derivatives of NCCs (Figure 4A); these malformations are rescued by coinjection of *hmga2* mRNA [39]. These alterations are anticipated, at neurula stage, by downregulation of key genes for neural crest specification (neural crest specifiers, NCS), like *snai2* and *twist* (Figure 4B); and, at tailbud stage, by downregulation of markers of NCCs in the migratory phase, like *twist*, *sox9*, *sox10*, *dlx2*, *tfap2* (in its late phase of expression) (Figure 4C,D). On the contrary, earlier genes that act at early neurula stage upstream of the NCS and serve to identify the neural plate border (neural border specifiers, NBS), like *msx1*, *pax3*, *tfap**2* (in its early phase of expression), are not affected by Hmga2 depletion. Further experiments also showed that MO-hmga2 was able to block the downstream effects of *msx1*, *pax3*, and *snai2* overexpression, suggesting Hmga2 requirement for the sequential action of these three genes [39].

Because part of the role of NCS genes, such as *snai1/2* and *twist*, is to elicit the EMT in NCCs [142], among the consequences of Hmga2 depletion there is the impairment of EMT and NCC migration: EMT-promoting genes like *snai2*, *twist*, *zeb2,* and *adam13* were downregulated in morphant embryos [39].

Interestingly, developmentally regulated EMTs share many aspects with pathological EMTs of tumors [143,144,145]. It is important to note that the HMGA2 protein plays a similar role in tumors, where it concurs to promote EMT through the direct activation of *Snai1*/*2* and *Twist* and the downregulation of E-cadherin [146,147,148,149,150,151,152,153,154,155,156]. It is feasible that HMGA2 cooperates and is recruited with different transcription factors in sequential steps of the EMT genetic program, in force of its capacity to interact with so many molecular partners [23,157].

NCCs are induced at the border between the neural plate and epidermis; BMP, WNT, and FGF signaling pathways, take part in this event [158,159,160]. There is ample evidence of several signaling pathways in triggering *Hmga2* expression. For example, *Hmga2* transcription is activated in response to TGF-β or BMP signaling [146,147,161,162], to the WNT canonical pathway [163,164] and to the FGF pathway [165]. Therefore, it is very likely that these pathways are involved in the developmental regulation of *Hmga2* in NCCs.

Work on *Xenopus* has also highlighted the role of *hmga2* in heart development. Interfering with *hmga2* function, by a dominant-negative *hmga2-engrailed repressor* fusion construct or by antisense MOs, resulted in impairment of normal heart development: expression of the cardiac, specific transcription factor gene *nkx2.5* was downregulated from neurula stage. At tadpole stage, the morphant embryos showed reduced heart size and reduced or absent heart contractions [38]. These effects were rescued by coinjection of wild type *hmga2* constructs. It was also showed that both Hmga2 and Smad1/4 have binding sites on the frog *nkx2.5* promoter, some of which are conserved in humans, mouse, and chick. Experiments performed in cultured cells, showed activation of *nkx2.5* by the Smad1/4 complex; while *hmga2* overexpression did not seem to have effect per se, a strong synergistic action on the *nkx2.5* promoter was observed by Smad1/4 and Hmga2 coexpression. It was also demonstrated that Smad1/4 physically interacts with Hmga2. The interpretation of these results is that Hmga2 and Smad1/4 proteins bind to the regulatory region of *nkx2.5*, a key gene for heart development; binding to the Smad and Hmga2 target sites may be facilitated by their mutual interactions, therefore explaining their synergy on the *nkx2.5* promoter [38].

### 3.4. HMGA, Proliferation, and the Cell Cycle

As remarked in the first part of this review, *HMGA* genes have effects on the control of size and weight; these effects of *HMGA1/2* are linked to their action in regulating proliferation and the cell cycle. The *Hmga1*^−/−^/*Hmga2*^−/−^ mice showed a ‘*superpygmy*’ phenotype with smaller size than *pygmy Hmga2*-null mice, a significant decrease in body size, fat tissue and a reduced viability. The proliferative effects exerted by HMGA1 and HMGA2 may explain the phenotype observed in single *Hmga1* or *Hmga2* and in double *Hmga1/Hmga2* knockout mice. *Hmga1*^−/−^/*Hmga2*^−/−^ MEFs have a lower growth rate than wild type, *Hmga1^−/−^* or *Hmga2^−/−^* MEFs [58]. Compared to wild type, *Hmga1*^−/−^/*Hmga2*^−/−^ MEFs show higher levels of the cell cycle inhibitor p27; moreover, the activity of the transcription factor E2F1 was impaired in *Hmga1*^−/−^/*Hmga2*^−/−^ MEFs and tissues. The reduction in E2F1 activity and the consequent decline of expression of the E2F1-dependent genes involved in cell cycle progression (*Cyclin A* and *Cyclin E*) may account for the *superpygmy* phenotype of double knockouts [58]. In addition, the cell cycle inhibitors p16^INK4A^, p19ARF, and p21^CIP1/WAF1^ were increased and may also cooperate to prevent RB phosphorylation and lead to growth arrest [58].

A large number of in vitro studies carried out on normal and tumor cells demonstrated that *HMGA1*/*2* dysregulation causes alterations in cell cycle progression and cell proliferation. We will here provide a few examples of the functional interactions of HMGA proteins with regulators of cell cycle and cell proliferation. Much of this evidence has been gained from in vitro studies on cultured cells that are models for studies in cancer biology (Figure 3D).

Different transcription factors involved in cell proliferation, such as c-MYC, MYCN, SP1, and E2F1 are able to bind to the *HMGA1* promoter and regulate its expression [75,166,167,168]. For example, the *HMGA1* promoter contains an E2F binding site and three putative SP1 binding sites; in T98G glioblastoma cells, E2F1 binds to the *HMGA1* promoter and cooperates with SP1 to regulate *HMGA1* transcription [169]. Moreover, HMGA1 promotes E2F1 activity by displacing the histone deacetylase (HDAC) 1 from the pocket domain of pRB in the pRB/E2F1 complex. In this way, HMGA1 inhibits pRB and its function in cell cycle arrest and allows E2F-mediated transcription. Therefore, the transcriptional repression of *Cyclin E*, a positive regulator of the G1/S transition, is removed, and E2F1 can bind to the *Cyclin E* promoter. In this way, HMGA1 exerts a positive effect on G1/S transition and on proliferation [168].

Overexpression of *HMGA1* in MCF-7 breast cancer cells increases proliferation, while its downregulation has the opposite effect; in these cells, *HMGA1* upregulates several cell cycle genes like *CLK-1*, *Cdc25A*, *Cdc25B*, *Cyclin C*, *JNK2*, and *MAPK* [170]. It was also found that *HMGA1* increases the percentage of cells in S phase, while it reduces G0/G1 cell population, and that transfected cells enter the S phase earlier than normal, causing an abnormally long G2-M phase [171].

In cervical cancer, *HMGA1* accelerated the G1/S phase transition by upregulating the expression of *Cyclin D1* and *Cyclin E1*, as demonstrated by in vitro and in vivo experiments in which cancer cells were subcutaneously injected into nude mice [172].

*HMGA1* also plays a role in lymphoid proliferation/differentiation by upregulating interleukin 2 (IL-2) and downregulating interleukin 6 (IL-6) [173]. Transgenic mice overexpressing *Hmga1a* develop T-cell acute lymphoblastic leukemia/lymphoma phenotype (T-ALL) [174]; the effect of *Hmga1a* overexpression on proliferation is exacerbated by the loss of the *Cdkn2a* tumor suppressor gene, that encodes, from alternative reading frames, the two cell cycle inhibitors p16^INK4A^ and p14^ARF^ (p19^ARF^ in mice) [175].

A time course transcriptomic analysis was performed by comparing lymphoid cells from wild type mice with cells from transgenic mice, that develop lymphoid leukemia as a consequence of *Hmga1* overexpression. This study found that among upregulated genes are *Cyclin A1* and *A2*, *Cyclin B1*, *Cyclin E*, further suggesting that *Hmga1* can drive a cascade maintaining cell cycle progression and proliferation [176].

On the other hand, negative regulation of *HMGA1* expression generally has opposite effects. HMGA1 expression is regulated by some small non-coding RNAs that impinge on cell cycle/cell proliferation. For example, miR-625 suppresses cell proliferation and migration of breast cancer cells by targeting the 3′-UTR of *HMGA1 mRNA;* consistent with this, overexpression of *HMGA1* recovers the miR-625-mediated inhibition of cell proliferation [177].

Furthermore, *HMGA1a* and *HMGA1b* are targets of miR-16, that is involved in cell proliferation and apoptosis [178]. miR-16 also targets *Caprin-1* (cytoplasmic activation/proliferation-associated protein-1) and *Cyclin E* in MCF-7 and HeLa cancer cell lines [178]. All these proteins are involved in lymphocyte differentiation/proliferation pathways, as well as in neoplastic processes [11,173,179]; miR-16 may therefore coordinately act on their mRNAs to impact on cell cycle and prevent proliferation.

Similar to *HMGA1*, *HMGA2* is also involved in cell cycle regulation in normal and cancer cells, and its dysregulation has been shown to alter the normal cell cycle progression, possibly leading to cell cycle arrest and apoptosis (Figure 3D).

*HMGA2* overexpression in transgenic mice induced proliferative hematopoiesis and clonal expansion of hematopoietic stem cells (HSCs), with increase of pStat5 and pAKT levels [175,180]. *HMGA2* is involved in both proliferation and differentiation of human hematopoietic stem and progenitor cells [181].

In different cell lines, HMGA2 activates the transcription of *Cyclin A*, a key factor that regulates S phase progression and G2/M transition. HMGA2 interferes with the binding of the repressor p120^E4F^ to the *Cyclin A* promoter, therefore favoring *Cyclin A* transcription, possibly by recruiting positive regulatory factors onto the c-AMP responsive element (CRE), necessary for full activation of the *Cyclin A* gene [182].

Like HMGA1, also HMGA2 is able to displace HDAC from pRB by binding to the pocket domain of pRB, thus promoting the release of pRB from E2F1, the acetylation of E2F1 and of the histones at the target sites, in this way allowing full activation of E2F1 and transcription of its target genes [59,183].

The positive action of HMGA2 in promoting cell cycle progression and proliferation is potentiated by impairment of cell cycle inhibiting mechanisms. When *p27^Kip1^* is deleted, or CDK4 responsiveness to INK4 inhibition is disrupted by a mutant CDK4, the effects of *Hmga2* overexpression are more severe: in models of pituitary adenomas, mice develop tumors much earlier, with a higher proliferative rate and a much higher number of mitoses (as measured by Ki67 detection) [184].

HMGA2 mediates the proliferative response of tumor cells to WNT/-CATENIN signaling pathway in a *Wnt10b*-driven breast cancer model. The *HMGA2* promoter contains WNT responsive elements that are shown by ChIP to be bound by β-CATENIN; in response to WNT10B, *HMGA2* is activated, and in turn two of its targets, the cyclin genes *CCNA2* and *CCNB2*, are upregulated [163]. The response to WNT10B is abolished by interfering with *HMGA2* function in Wnt10b-driven tumor cells. Interfering with WNT signaling or with *HMGA2* also reduces proliferation in MDA-MB-231 cells, but proliferation is resumed by *HMGA2* overexpression (under persisting WNT inhibition), suggesting that *HMGA2* acts downstream of WNT10B [163]. Interestingly, during mouse development, HMGA2 protein is detected at E14.5 in the developing mammary placodes of wildtype mice, but not of *Wnt10*
^−/−^*b* mutants. These data point towards the existence of a WNT10B/β-CATENIN/HMGA2 regulatory axis, for which also the chromatin remodeling factor EZH2 seems essential [164].

An increase in the expression of HMGA2, and of its target IGF2BP2, was detected in women with polycystic ovary syndrome. The activation of the HMGA2-IGF2BP2 pathway upregulated the expression of *Cyclin D2* and of *SERBP1* (a gene encoding *SERPINE1* mRNA binding protein 1) in granulosa-like tumor cell lines; this upregulation in turn led to increased cell proliferation [185].

On the contrary, negative regulation of HMGA2, usually causes diminished proliferation. For example, in lung cancer cells, the upregulation of miR-495 repressed HMGA2 expression by binding to its 3′-UTR and inhibited cell proliferation; HMGA2 upregulation reversed this inhibition [186]. In bladder cancer cells, silencing of *HMGA2* suppresses cell proliferation [187].

Therefore, while *HMGA1*/*2* overexpression tends to promote cell cycle progression and proliferation, the impairment of *HMGA* function has the opposite effect, leading to lower proliferation; this impairment often results in cell cycle arrest, senescence, and apoptosis (Figure 3D).

For example, under serum starvation, T98G glioblastoma cells undergo growth arrest in the G0 phase, through a RB-mediated mechanism. This G0 phase is characterized by reduced general transcription, reduced phosphorylation of RB and reduced amount of HMGA1. HMGA1 may be relevant in this context, as its overexpression is able to withdraw cells from this G0 arrest, suggesting that downregulation of HMGA1 is mechanistically essential for G0 arrest [168]. It was also shown that HMGA1 itself is a target of E2F1 [169].

In cells of neurofibromatosis type 1-associated malignant peripheral nerve sheath tumors, *HMGA2* knock-down impaired cell cycle, led to a reduction in the percentage of dividing cells and to arrest in G0/G1 phase; in addition, there was an increase of apoptotic cells, an increased BAX1 expression, and a decrease in the expression of BCL-2 and of Cyclin D1 (related to the G0/G1 phase), suggesting that knock-down of *HMGA2* function leads to apoptotic cell death [188]. In breast cancer cells, *HMGA2* downregulation induced by miR-98 inhibited cell proliferation and caused apoptosis [189]. In cervical cancer cells siRNA-induced *HMGA2* downregulation inhibited cell proliferation and promoted apoptosis, showing a decreased *BCL-2* expression and increased *CASP 3* expression; on the contrary, *HMGA2* overexpression caused opposite effects [190].

In conclusion, although the molecular details are not completely worked out, there is a good deal of evidence that HMGA proteins promote cell cycle progression and sustain cell proliferation, while impairing HMGA function leads to a lower proliferation, or to cell cycle arrest, senescence and apoptosis (Figure 3D).

However, at variance with the overall picture depicted above, HMGA proteins are sometimes enrolled in promoting senescence, apoptosis, and chromosomal instability.

Cellular senescence represents a state of stable cell cycle arrest [191]. Senescence is a heterogeneous condition, and its phenotype depends on the cell type of origin and on the initial triggering event. For example, senescence can be accompanied by deep and visible alterations of the chromatin structure, such as the senescence-associated heterochromatic foci (SAHFs); or by a change in the cell secretory activities (senescence associated secretory phenotype, SASP) [192,193,194].

Recent work has shown that HMGA proteins are structural components of SAHFs formed in RAS-induced IMR90 senescent cells. In these cells, HMGA1/2 proteins contribute, together with the p16^INK4a^ tumor suppressor, to SAHF formation, and are required for cell cycle arrest through their negative regulation of proliferative genes [194]. HMGA depletion abolishes SAHF structure at all stages, whereas p16^INK4A^ is required only at initial stages of SAHF formation [194]. Notably, the co-reduction of p16^INK4A^ and HMGA1 allows cells to escape senescence, demonstrating that both proteins contribute to senescence cell arrest [194,195].

*HMGA* genes seem to be differently involved in RAS- and NOTCH-induced senescence. In the first case, both HMGA proteins are upregulated and involved in chromatin modifications accompanied by SAHF formation; in the second case, *HMGA1* and *HMGA2* expression is instead suppressed, and overall changes in chromatin structure are detected without SAHF formation. An analysis of chromatin accessibility revealed that RAS induced an open and accessible chromatin state at many genomic regions of IMR90 cells, including *HMGA1* enhancer locations (consistent with HMGA1 protein being active). NOTCH signaling could repress this accessible state at *HMGA1* enhancer regions, and presumably shut off *HMGA1* expression, while it was not possible to detect similar modifications at the *HMGA2* gene. It also turned out that NOTCH was also able to repress chromatin accessibility at most of the sites opened up in RAS-induced senescent cells, and that this effect of NOTCH was very likely due to the suppression of HMGA1 expression [196].

Therefore, the regulation and role of HMGA in senescence appears to be context-dependent.

HMGA2 downregulation, associated with increased expression of senescence proteins p16^INK4A^, p21^CIP1/WAF1^, and p27^KIP1^, was detected in human umbilical cord blood-derived multipotent stem cells induced to senescence by inhibition of HDAC activity. In this case, different miRNAs (including let-7 family members, miR-23a, miR-26a, and miR-30a) were upregulated and targeted *HMGA2* mRNA for negative regulation [197]. On the contrary, *HMGA2* overexpression suppressed p16^INK4A^ and p21^CIP1/WAF1^ through the PI3K/Akt/mTOR/p70S6K pathway, and reversed senescence [198] (Figure 3D).

In a study on ovarian cancer, let‑7d‑5p was shown to target *HMGA1* mRNA and to activate the p53 pathway, leading to impairment of cell cycle progression and proliferation, and to apoptosis. let‑7d‑5p induced increased levels of p21, Bax, p27, and p53wt, while HMGA1, p53mut, PCNA, CDK2, MMP2, MMP9, and BCL‑2 were downregulated [199].

On the other hand, there are evidences that, in some contexts, HMGA1 and HMGA2 may actively promote apoptosis. For example, HMGA2 overexpression in primary human fibroblasts caused upregulation of the pro-apoptotic markers p53, BAX, and cleaved caspase 9, downregulation of the anti-apoptotic marker BCL-2 and release of cytochrome c from the mitochondria; in addition, there was also concentration of H2A in nuclei and increase of caspase-2 level, suggesting that overexpression of HMGA2 may promote DNA damage, thus initiating caspase 2 activation and cytochrome c release, that in turn leads to progression of the apoptotic cascade [200].

### 3.5. HMGA in EMT

Hmga2 participates in the gene regulatory program of NCCs in *Xenopus*, and its downregulation by MO injections produces severe disruption in craniofacial derivatives, possibly due to mis-specification of NCCs and to impairment of EMT and migration [39]. On the contrary, we did not find direct evidence in the literature for a role for HMGA1 in NCCs, although some neural crest derived tumors do express HMGA1 [167,201,202]. Both HMGA proteins, however, have been involved in EMT processes in a variety of tumor cells (Figure 3C).

HMGA1 promotes EMT in different types of cancer cells [166,203,204]. As an example, overexpression of HMGA1 leads to EMT in basal-like breast cancer cell [203]. Interestingly, HMGA1 is a direct target of miRNAs regulating EMT and migration in different cancer types. In particular, miR-4458 inhibits the process of migration and EMT of non-small cell lung cancer via targeting HMGA1 [205]. In addition, in a lung adenocarcinoma cell line, the overexpression of miR-26a inhibits cell migration and invasion by targeting HMGA1 [206]; miR-625 represses proliferation and migration by targeting HMGA1 in breast cancer cells [177]. HMGA1 promotes migration in breast cancer and is targeted by let-7 [207], while it promotes migration of bladder cancer cells (also inhibited by let-7) [208] and of hepatocellular carcinoma cells [209] (Figure 3C).

HMGA1 overexpression can promote endothelial-to-mesenchymal transition (EndoMT). Indeed, it was shown that the reduced expression of BMPR2 in pulmonary arterial endothelial cells causes an upregulation of HMGA1 expression, that in turn promotes an increase of Endo-MT markers (SM actin, SM22, calponin, phospho-vimentin, SNAI2) and the reduction of the platelet endothelial cell adhesion molecule CD31; the cells showed a smooth muscle-like morphology, similar to cells derived from patients affected by pulmonary arterial hypertension [210] (Figure 3C).

HMGA2 involvement in EMT is very well documented in a variety of tumors, and we have discussed some of its involvement in tumor EMT and migration in connection with NCCs [146,147,150,151,152,153,154,211,212,213,214,215,216,217,218,219]. HMGA2 is also involved in EMT of crystalline epithelial cells in lens fibrosis; this process apparently uses the same pathway of TGFβ-regulated EMT, recruiting SMAD3, HMGA2, and SNAI [220,221,222].

### 3.6. HMGA in Stemness Maintenance

Together with the role exerted by HMGA proteins in cell cycle progression and cellular proliferation, growing evidences concerning their role in stemness and self-renewing maintenance have been reported.

HMGA1 was described as a positive regulator of pluripotency in induced pluripotent stem cells (iPSCs) and human embryonic stem cells (hESCs). High levels of HMGA1 expression are observed in these cells. When HMGA1 is overexpressed during somatic cell reprogramming together with OCT4, SOX2, KLF4, and C-MYC factors, the efficiency of reprogramming is enhanced, and the expression of genes involved in maintaining pluripotency is increased; on the contrary, in hESCs knock-down of HMGA1 fosters differentiation and the repression of the pluripotency genes. Accordingly, HMGA1 binding sites have been demonstrated in *SOX2*, *LIN28* and *c-MYC* promoters by ChIP, whereas putative HMGA1 binding sites have been predicted in the promoters of the stem cell maintainance genes *OCT4* and *NANOG* [223,224]. Interestingly, in combination with SOX2, HMGA2 was shown to promote reprogramming of human somatic cells into NSCs, with a higher efficiency than SOX2 alone or different combinations [225]. As outlined above, HMGA proteins are essential at early developmental stages for the regulation of the neurogenic potential of NPCs during early neocortical development, through induction of an open chromatin state [120].

Other evidence of the role of HMGA1 and HMGA2 in maintaining a stem cell state has been found in different types of tumor cells, among which we will only provide a few examples (Figure 3B).

Evidence for the HMGA role in maintenance of neural stemness comes from tumors of neural origin, particularly from glioblastoma multiforme (GBM). HMGA1 expression was related to the proliferative and invasive properties of GBM [226]. In glioblastoma cells HMGA1 maintain stemness by modifying the chromatin status and favoring transcription of stemness promoting genes. A great part of this effect seems mediated by *SOX2*, whose locus is actively remodeled by HMGA1 through displacement of histone H1; on the other hand, miR-296-5p targets HMGA1 mRNA to inhibit stemness [227].

*HMGA2* is also involved in glioblastoma stem cell maintenance. In this context, an important part of *HMGA2* action takes place through the LIN28A-let-7-HMGA2 axis, working in a similar way as in normal neural development. LIN28A and HMGA2 sustain stemness, growth and invasivity of GBM cells in vitro and in vivo, while let-7 has the opposite effect by negatively modulating *HMGA2* mRNA translation [228].

HMGA2 is a positive regulator of the stemness-maintaining gene *SOX2*, and, similar to HMGA1, binds to the *SOX2* promoter. Like HMGA1, HMGA2 is also negatively regulated by miRNAs; in this case, miR-142-3p acts on the 3′UTR of *HMGA2* mRNA and inhibits its translation, in turn leading to *SOX2* downregulation and stemness inhibition; IL-6 also plays a role in this circuit by epigenetically suppressing miR-142-3p expression [123].

Another important player on which HMGA2 positively acts is *FOXM1* [125], encoding a transcriptional regulator that promotes stemness in mouse NSCs and in different tumors [229,230,231,232,233].

In the endodermal layer, *Hmga1* plays a role in creating a stem cell niche required for the homeostasis of the epithelial lining of the intestine and for the maintenance of intestinal stem cells (ISCs), where *Hmga1* is highly expressed [234]. In the intestine of transgenic mice, overexpression of *Hmga1* leads to hyperproliferation and to the development of abnormal crypts and polyps; furthermore, *HMGA1* causes metastatic progression and endows colon cancer cells with stem cell-like abilities [30,235]. Consistent with this, in colon tumor stem cells (CTSCs), HMGA1 is expressed at higher levels than in normal and in colon cancer cells. In this context, HMGA1 amplifies the WNT/β-CATENIN signaling and favors self-renewal and increase of the ISC compartment [30,234]. Crypt cultures from *Hmga1* transgenic mice showed an increased ability to give rise to organoids, with enhanced bud formation, compared to control cultures, demonstrating an enhanced stem cell function. On the contrary, *Hmga1* repression in CTSCs increases stem cell quiescence, decreases self-renewal and alters the three-dimensional organization of organoids and of buds, as well as that of colonosphere [234,236]. In addition, *Hmga1* also helps to arrange the intestinal stem cell (ISC) niche by promoting terminal differentiation of Paneth cells. Paneth cells secrete WNT3A and other signals that support self-renewal and maintenance of ISCs in the crypts. *Hmga1* is able to expand the Paneth cell compartment by upregulating *Sox9*, a WNT target gene essential for Paneth cell differentiation [234]. *HMGA1* itself is a target of WNT/β-CATENIN signaling, so a positive feedback between these two molecular players may initially serve to start and then maintain the ISCs compartment [237,238,239,240].

Other evidence from colorectal cancer cells suggests that HMGA1 promotes their self-renewal ability by enhancing the expression of stemness markers (CD44, CD133, OCT4, ABCG2, ALDH1, and BMI-1), and acting on the PI3K/Akt and MEK/ERK pathways, leading to an expansion of the stem cell pool [241].

In breast cancer, both HMGA1 and HMGA2 play a role in maintaining stem cell properties to the tumor cells. HMGA1 is involved in reprogramming cancer cells to a stem-like status and in favoring metastatic progression in triple negative breast cancer cells (MDA-MB-231 and Hs578T); silencing *HMGA1* inhibits stem cell properties within mammospheres and decreases invasive abilities of cells [203,204,223]. HMGA2 knock-down decreases NANOG, OCT4 and SOX2 expression; miR-33b targets *HMGA2*, *SALL4*, and *TWIST*1 mRNAs, acting as a negative modulator of stemness and invasion [242].

Other examples of involvement of HMGA proteins in stem cell state maintenance come from other tumor cell lines, like ovary cancer cells, oral squamous carcinoma cells, prostate cancer cells, and osteosarcoma cells. In these cell lines HMGA1 and HMGA2 positively regulate stem cell genes like *SOX2, KLF4, OCT4,* and *NANOG*, sometimes in a reciprocal loop; in several instances, the LIN28-let7-HMGA2 axis is recruited for the positive regulation of stemness genes [243,244,245,246].

One of the most important aspects of stem cell biology during development is the shift from a symmetric cell division (typical of early stem cells and cancer stem cells) to an asymmetric cell division (typical of later, non-pathological, stem cells). HMGA proteins may also influence the mode of cell division, acting on the regulation of NUMB expression (Figure 3B). NUMB is fundamental in the balance between symmetric and asymmetric divisions in stem cells, and it is lost/reduced in many tumors including GBM; *NUMB* mRNA translation is regulated by the mRNA-binding protein MUSASHI-1 (MSI1) and by the micro-RNA miR-146a. HMGA1 binds to the *NUMB* promoter and negatively regulates its transcription in GBM [247]; HMGA1 also counteracts *NUMB* expression at the post-transcriptional level, by upregulating MSI1 and miR-146a, both of which inhibit *NUMB* mRNA translation [247,248]. Noteworthy, NUMB exerts its major effect for maintaining normal neural and glioma stem cell state by inhibiting NOTCH1 signaling, which allows brain tumor growth and glioma stem cell proliferation [249,250,251]. Therefore, NUMB upregulation induced by HMGA1 depletion in turn leads to the downregulation of the NOTCH1 pathway, and to the asymmetric mode of cell division typical of non-pathological cells [226,236,247].

### 3.7. Other Developmental Effects of HMGA

HMGA proteins are involved in regulating proliferation/differentiation in the blood compartment (Figure 3E). Functional studies in both Jurkat T cells and primary T cells show that HMGA1 can bind to numerous sites across the *IL-2* proximal promoter region and modulate the binding of major transcription factors that control *IL-2* gene transcription, like AP-1 and NF-AT, thus regulating T cell proliferation [252]. In addition, HMGA1 protein participates in inducing the expression of the human *IL-2Ra* gene in response to T cell activation, by acting both on an upstream enhancer and on the proximal promoter element [253]. *Hmga1*^−/−^ ES cells generate fewer T-cell precursors than do wild-type ES cells; indeed, they preferentially differentiate into B cells, probably consequent to decreased *IL-2* and increased *IL-6* expression [173].

Ablation of HMGA1 expression also induces changes in myeloid lineage differentiation (such as the proportion of monocyte/macrophage precursors vs. megakaryocyte precursors), in erythropoiesis, and in gene expression. HMGA1 effects could be mediated by the GATA-1 transcription factor, a regulator of erythrocyte cell commitment and differentiation (Figure 3E). *GATA-1* is overexpressed in *Hmga1*^−/−^ ES cells. HMGA1 protein binds to the regulatory sequence of *GATA-1* for a negative modulation of its transcription, as shown by reporter assays in ES cells; transfection of the *Hmga1* gene in *Hmga1*^−/−^ ES cells restores normal levels of *GATA-1* reporter activation. In addition, in the lymphoid lineage, *Hmga1*^−/−^ ES cells showed a reduced ability to generate T-cell precursors compared to wild-type ES cells, and an increased ability to generate B-lymphocyte precursors [173].

HMGA2 plays a role in the blood compartment, within the LIN28-let7-HMGA2 interplay [254,255,256] (Figure 3E). In the mouse, LIN28B expression in myeloid progenitors decreases during hematopoietic maturation from the fetal stage up to the adult stage. This decrease is paralleled by the increase of let-7. When let-7 is downregulated in the adult hematopoietic system, a fetal erythroid-dominant hematopoiesis is favored; conversely, absence of *Lin28b* or activation of *let-7* in the fetal state leads to an adult-like myeloid-dominant hematopoietic scheme. HMGA2 stands as a let-7 modulated effector of this switch in commitment/differentiation [254,256]. Interestingly, there is a timely aspect in this process that closely recalls the let-7/HMGA interplay taking place in neocortical histogenesis [44,51].

In human chondrocytes, HMGA1 positively regulates the transcription of the *insulin-like growth factor-binding protein 3 (IGFBP-3)* gene, by directly interacting with its promoter [257]. IGFBP-3 regulates the level of the insulin-like growth factor IGF-1 that is important for chondrocyte survival and for cartilage maintenance. In osteoarthritis (OA), chondrocytes do not respond properly to IGF-1, possibly because of higher IGFBP levels found in OA chondrocytes. Based on the observation that HMGA1 upregulation in human chondrocytes is associated with increased expression of IGFBP-3, it has been hypothesized that HMGA1-mediated IGFBP-3 overexpression may be responsible for the impaired response to IGF-1 signaling in OA [257].

HMGA1 cooperates with the serine-threonine kinase homeodomain-interacting protein HIPK2 in regulating lung and thyroid development. HIPK2 is able to phosphorylate HMGA1, thus decreasing its binding affinity to DNA and altering the HMGA1-mediated regulation of transcriptional activity [258]. The *Hmga1*/*Hipk2* double knock-out mice show an altered lung morphology, possibly caused by the diminished production of surfactant proteins (SP-A, SP-B, and SP-C) necessary for correct lung development [259]. Consistently, HMGA1 and HIPK2 proteins were known to positively regulate the expression of surfactant proteins. About 50% of the *Hmga1*/*Hipk2* double knock-out mice die for respiratory failure. In addition, these mice display an altered thyroid differentiation, most probably due to a strong reduction in the expression of the thyroid specific transcription factors PAX8 and FOXE1 [259].

HMGA1 exhibits pro-angiogenic properties; in fact, HMGA1 knock-down reduced the expression of the pro-angiogenic factors ANGPT-1 and VEGFA expression in rat brain microvascular endothelial cells. Besides, HMGA1 depletion decreased cell viability in these brain endothelial cells [260]. More recently, HMGA1 was shown to interact with the FOXM1 transcription factor for VEGFA promoter binding, inducing tumor angiogenesis in HUVEC cells and in zebrafish xenografts [261]. Similarly, there are also evidences of HMGA2 being involved in the regulation of pro-angiogenic genes like VEGFA, VEGFC, and FGF-2 [262] and in the angiogenesis occurring in diabetic retinopathy [263] (Figure 3F).

HMGA1 is involved in lens differentiation, where it interacts with BRG1 to promote *CRYAB* expression (Figure 3F). The CRYAB protein is highly expressed in vertebrate eye lens and has been associated with diseases involving deficiencies of the crystallin proteins, such as cataract disease. In vitro and in vivo experiments demonstrated that HMGA1 binds to an AT-rich sequence within a BRG1-responsive element located upstream of the *CRYAB* transcription start site and is required for the BRG1-mediated activation of the gene [264].

HMGA1 deficiency causes insulin resistance and diabetes through impairing the transcription of the insulin receptor gene INSR and of the insulin-like growth factor binding protein-1 (IGFBP-1) genes [265,266,267,268,269]. Notably, the pancreatic islets from mutant animals appear smaller and irregular when compared with wild-type islets, indicating that HMGA1 downregulation may cause problems during differentiation, leading to histological and functional alterations of pancreatic tissue [265]. On the other hand, evidence obtained both from cultured hepatic cells and in vivo showed that the histone methyltransferase G9a/EHMT2, an important regulator of insulin signaling, acts through a HMGA1-dependent mechanism to rescue insulin signaling in *db*/*db* mice, a mouse model for type 2 diabetes mellitus [270]. Altogether, these results strongly suggest that HMGA1 deficiencies may contribute to the development of specific forms of diabetes mellitus.

## 4. HMGA and Evolution

*Hmga* genes are present throughout the animal kingdom and also in plants [18,271]. A comparison of deuterostome Hmga protein sequences led to the possible conclusion that chordate ancestors had only one *Hmga* gene, and this was duplicated in the Gnathostome lineage: in fact, all vertebrates seem to have both Hmga1 and Hmga*2*, with the possible exception of Agnathans and, curiously, of *Xenopus*; the case of *Xenopus* may be peculiar, given that other Amphibians have both Hmga [272]. The phylogenomic analysis of deuterostome Hmga, using Cnidarian Hmga-like sequences as outgroup, suggests that a single Hmga was present in the closest chordate ancestors; this Hmga might have been closer to Hmga2 than to Hmga1. In fact, both the Hmga-like sequence of amphioxus (*Branchiostoma* spp., and *Asymmetron lucayanum*) and that of *Saccoglossus kowalewskii* share the typical W83 residue after the third AT-hook, present in all Hmga2 sequences but absent in Hmga1. In addition, both for the number of exons and the genomic landscape around the locus, amphioxus *hmga* looks definitely closer to human *Hmga2* than to *Hmga1* [272]. Other deuterostome Hmga-like sequences (e.g., Echinoderms) do not have the W83 residue and cluster separately from the amphioxus and *Saccoglossus* sequences as well as from vertebrate Hmga1 and Hmga2 [272].

It might be interesting to note that in amphioxus, the *hmga*-like sequence is expressed in the endoderm around the gill slits, recalling *hmga2* expression in the pharyngeal arches of *Xenopus*, populated by the NCCs that make up the branchial skeleton. Recent work has suggested that a proto-chondrogenic set of genes (like *Sox*, *Runt*, and *Hedgehog*) was present in chordate ancestors and active in the notochord, neural tube, and in the gill mesoendoderm of protochordates. This set of genes would have been co-opted by vertebrate skeletogenic NCCs to build the pharyngeal branchial skeleton [273,274,275]. This is certainly suggestive, given that the collagen skeleton of the pharyngeal region of amphioxus is secreted by endodermal cells [276] and given the similarity of expression of *BfHMGA* in amphioxus to that of *Hmga2*. Interestingly, HMGA2 was found to enhance chondrocyte proliferation [277,278]. It may be tempting to speculate that the ancestral chordate Hmga may also be part of a chondrogenic set of genes involved in constructing a proper pharyngeal skeleton.

Direct and indirect evidence is consistent with the involvement of Hmga2 in craniofacial development and evolution. Recent work has pinpointed the genomic region containing the *HMGA2* gene as associated to the phenotypic variation occurring in beak size among Darwin’s finches. Changing environmental conditions may expose these birds to strong selective events that are constantly monitored on the field [279,280]. In severe droughts that occurred in past years, the population of medium ground finches entered severe competition with large ground finches for food availability; medium ground finches that reduced their beak dimensions preferentially survived with respect of cospecific individuals with larger beaks. By genotyping the different individuals, two independent studies identified alternative haplotypes, respectively favoring large or small beak dimensions; SNPs were found in the *HMGA2* genomic region and this gene was indicated as the prime candidate to mediate this phenotypic variation because of its described effects on organ and body size; this phenotypic variation may potentially lead to ecological segregation and eventually to speciation [279,281,282,283]. Beak development is strictly connected to NCCs [141] and therefore direct support for an involvement of *HMGA2* in craniofacial development and evolution comes from the aforementioned work on *Xenopus* [39], and from the observations of craniofacial disturbances in the rabbit *dwarf* mutant [87] and in the *hmga2*^−/−^ pygmy mouse [53]; indirect support comes from studies that suggest the *HMGA2* gene to be associated to craniofacial size and shape [284,285,286,287,288].

## 5. Conclusions

HMGA proteins, with their chromatin remodeling abilities, are involved in modulating many processes in cell biology and development. Because they are able to modify the chromosome architecture and give access to the DNA to many regulatory factors, their effects are expected to be pleiotropic and therefore may concern many genetic networks. We expect that in the future new findings will expand our knowledge on HMGA mechanisms of action in the processes in which they are already known to participate, or find new roles for them in development. The overview given here is mainly seen from a developmental biology standpoint but has needed to consider information from cancer and cell biology in order to show the multifaceted contribution of these proteins to development.

There are some recurrent themes occurring in the biological functioning of HMGA. One of them deals with their role in sustaining cell cycle progression by facilitating the expression of key genes of the cell cycle machinery. In this respect, most evidence discussed here leads to interpret their action as that of cell proliferation driving genes; this may be a partial interpretation, given their involvement also in processes that apparently have an opposite output, like cell arrest and senescence or even cell death. Therefore, beyond cell cycle and proliferation, a perhaps better reading is that of a complex and integrated function, where HMGA are “hubs” that participate to different aspects of cell biology and development that need to be balanced and coordinated: from the cell cycle (cell proliferation vs. cell senescence vs. cell death), to cell commitment (cell renewal vs. differentiation), to setting the time for a particular developmental event to occur/start/finish (EMT and migration; timely generation of neurons or glia, or of different blood cell types). As “hubs” [18] they can influence many processes in different cellular contexts, under different regulatory inputs (Figure 3). Certainly, one of the most interesting aspects concerns the meaning of their post-translational modifications; in fact, while some of these modifications were suggested to modulate HMGA activity in cell cycle and apoptosis [16,27,258,289], to our knowledge much less is known about their causal significance in terms of commitment, differentiation, and cell movements.

There are also recurrent regulatory axes to which they participate and certainly the LIN28-let-7-HMGA is a prominent one. This seems a very potent pathway [28] into which HMGA2, and sometimes HMGA1, is engaged; it may serve to coordinate some of the functions that HMGA are involved in. In particular, we refer to proliferation, potency, commitment and differentiation, for which let-7 may set the pace and time by modulating HMGA protein levels, generally declining as let-7 goes up. This control mechanism is shared with other miRNAs, so that this class of molecules exert a strict control on HMGA, generally shutting off their action. This in turn leads to different outputs. For example, restricting the potency that cells seem to maintain under continued HMGA expression. Altering the let-7-HMGA interplay leads to a heterochronic shift in layering of the neocortex [44,51,120], or in generation of blood cell types [254,256]. There may be other developmental events where perturbations of this axis may produce similar heterochronic phenotypes: for example, the LIN28-let-7 interplay has been recently involved in NCC multipotency [290] and in tail bud extension [291], but in these cases HMGA expression was not examined; given that HMGA2 is expressed in NCCs, and given its interplay with LIN-28 and let-7, it would not be surprising if HMGA2 also participates in determining NCC potency; on the other hand, anti-HMGA2 MO injections targeting the posterior part of the *Xenopus* embryo produce tail reduction [38], suggesting that HMGA2 may also be active in tailbud extension.

HMGA1 and HMGA2 are involved in the EMT process, both during development (e.g., in NCCs) and in tumor progression. In both cases the molecular machinery recruited for EMT is very similar and involves the same key regulators. NCCs are the main cell population involved in building the craniofacial skeleton and in giving a species-specific appearance to the face [141]. As outlined above, involvement of HMGA2 in this process comes from direct and indirect evidence. Recent work has shown that chromatin remodeling factors play an important part in the genetic program that specify NCCs and promote their EMT. The chromatin remodeling factor CHD7 cooperates with PBAF (polybromo- and Brg1- associated factor-containing complex) to remodel regulatory elements of the NCC key genes *TWIST* and *SOX9*; in the frog embryo, interfering with Chd7 (or Brg1 or Brd7) function leads to cranial skeletal defects [292]. Epigenomic profiling of the chromatin landscape in NCCs differentiated from hESCs has identified about 4300 human NCC enhancers with an active chromatin signature; it also identified a functional cooperation between TFAP2A, a master NCC gene, and NR2F1 and NR2F2 nuclear receptors in the NCC genetic program [293,294]. A list of about 1000 genes whose enhancers are co-bound by TFAP2A and NR2F1/2 was reported; many of these are known genes involved in anterior NCC development and in human craniofacial anomalies; notably, both HMGA1 and HMGA2 were listed [294].

It was shown recently that replacement of the canonical H2A with the H2A.Z.2 variant, is a necessary step for correct craniofacial morphology. SRCAP is an AT-hook containing remodeling factor that catalyzes this substitution; heterozygosity for a truncated allele at this locus leads to the Floating-Harbor syndrome, a developmental disorder with altered craniofacial features (sometimes with cleft lip) and other abnormalities, including bone growth delay, and diminished stature and weight during early childhood. The truncating allele removes the terminal part of SRCAP, containing the three AT-hooks motifs, and prevents the protein to enter the nucleus. Significantly, downregulation of SRCAP in the frog embryos leads to cranial dysmorphologies; downregulation of the H2A.Z.2 variant histone (but not of the close variant H2A.Z.1) mimics these dysmorphologies, while its overexpression (but not that of H2A.Z.1) rescues SRCAP ablation effects [295]. It was also observed that AT-rich sequence motifs were enriched at H2A.Z.2-biased enhancers, while GC-rich sequences were enriched at H2A.Z.1 bound enhancers, suggesting a possible dependence on the SRCAP AT-hook for the specific pattern of H2A.Z.2 distribution. It will be interesting to determine whether HMGA proteins have any cooperative role with remodelers of NCC enhancer regions, like CHD7, BRG-1, SRCAP, given that HMGA interact with some of them [264]. It is feasible that HMGA play a pleiotropic role in this context, by interacting with different transcription factors and/or by participating in some of the epigenetic modifications taking place at enhancer regions; some of these interplays and epigenetic regulations may change during evolution and these modifications may lead to morphological differences.

Beyond NCCs, the EMT process has been related to some “potency refreshment” that cells undergoing EMT may have in terms of stemness. The acquisition of a mesenchymal phenotype may be linked to a re-acquirement of stem cell-like properties, though this is still in debate [145,296,297,298]. Interestingly, some of the molecular steps involved in regulating stemness, EMT, and NCC biology see the participation of the LIN28-let-7-HMGA axis, so that these aspects may be integrated, also in the temporal dimensions, by a coordinated genetic mechanism, that finds one of its “hubs” in the HMGA proteins [18].

## Figures and Tables

**Figure 1 ijms-21-00654-f001:**
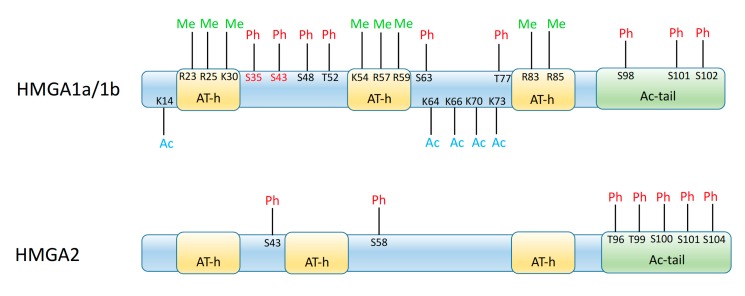
Schematics of the functional organization of HMGA1 and HMGA2 proteins, showing the AT-hook domains (AT-h) and the acidic terminal tail (Ac-tail) of the proteins, as well as the residues that may be modified by acetylation (Ac), phosphorylation (Ph), or methylation (Me). Numbers indicate the position of these residues in the mature protein (initial methionine is removed post-translationally); for the HMGA1a/HMGA1b scheme they refer to the HMGA1a sequence; residues in red are only present in HMGA1a and spliced out from HMGA1b, while those in black are present both in HMGA1a and HMGA1b.

**Figure 2 ijms-21-00654-f002:**
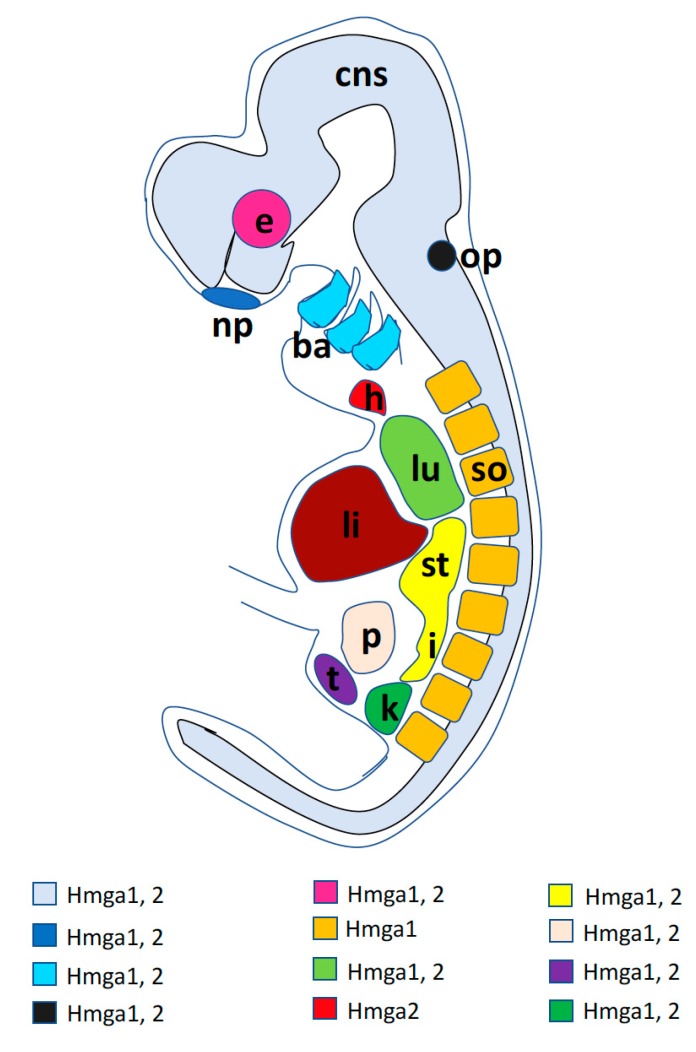
Graphic summary of the main sites of expression of *Hmga1* and *Hmga2* genes in a generalized vertebrate embryo. Colored codes resume the expression of the two genes as described in the text: ba, branchial arches; cns, central nervous system; e, eye; h, heart; i, intestine; k, kidney; li, liver; lu, lung; np, nasal placode; op, otic placode; p, pancreas; so, somites; st, stomach; t, testis.

**Figure 3 ijms-21-00654-f003:**
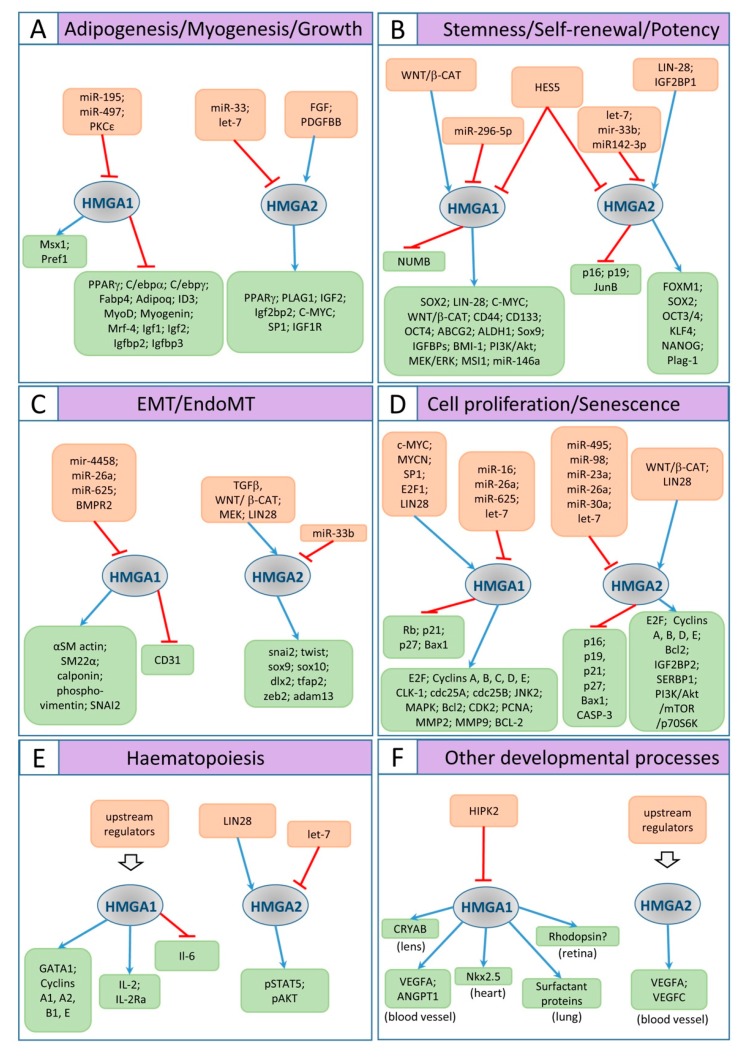
Schematic representation of the HMGA1/2 pathways involved in the cell and developmental biology processes described in the text. Orange boxes include regulators of HMG1/2 proteins; green boxes include molecules regulated by HMG1/2. Blue arrows indicate positive regulation; red bars point out negative regulation. (**A**) adipogenesis, myogenesis, and growth; (**B**) stemness/self-renewal/potency; (**C**) EMT and EndoMT; (**D**) cell proliferation and senescence; (**E**) hematopoiesis; (**F**) other developmental aspects.

**Figure 4 ijms-21-00654-f004:**
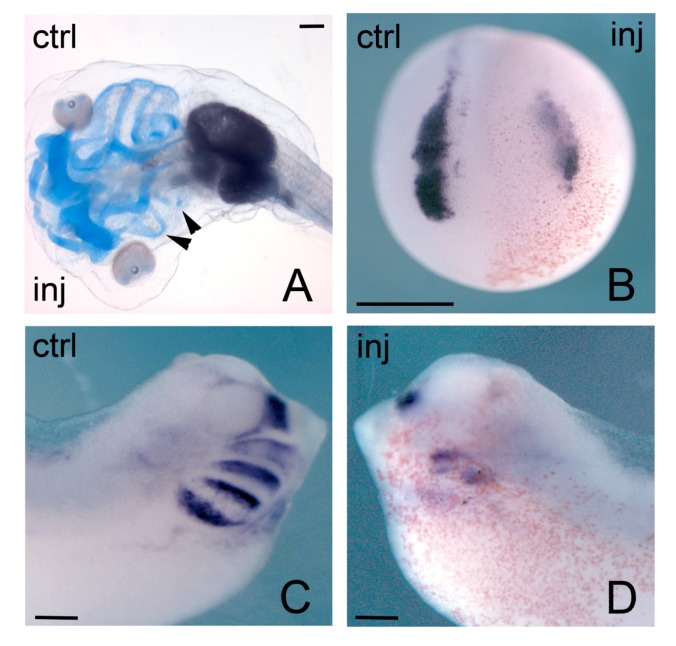
Phenotypic effects of Hmga2 depletion on *Xenopus* embryos. (**A**) Morpholino-injected stage 42 embryo show impairment of NCCs-derived skeletal elements on injected side as revealed by alcyan blue staining; (**B**) Neurula morphant embryo showing downregulation of *snai2* expression on injected side; (**C**,**D**) A clear reduction of *twist* expression was observed in injected side of stage 28 tailbud embryos when compared to control side. Inj, injected side; ctrl, control side. Scale bar: (**A**–**D**) 400 μm.

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
