# Peer review of "HMGA Genes and Proteins in Development and Evolution"

_ijms, 2020, doi:10.3390/ijms21020654_

Round 1

Reviewer 1 Report

In this manuscript Vignali and Marracci review the role of HMGA proteins in development and evolution. HMGA proteins are small nuclear proteins that can bind DNA and modify chromatin architecture and gene expression. Vignali and Marracci, after describing the developmental expression of the Hmga genes in different organisms, summarize the HMGA pathways involved in some cell and developmental biology processes. This review is very interesting and it is an important contribution to the field.

Although the submitted manuscript shows a complete overview of the research studies in the field, some small modifications would improve the quality of the work.

The focus of this review is on development and evolution, however a figure representing the structure of the HMGA proteins with their domains and post-translational modification sites would be appreciated in chapter 1. Is there something known about possible links between these modifications and the functions of these proteins described in chapter 3? If so, please include this information.

It would be appreciated if some figures (similar to Figure2) would guide the readers throughout the different parts of chapter 3. The figure 2 presented in the “5.conclusions” chapter contains many information discussed in previous chapters and could be more appropriate if splitted and presented as separate figures in the different parts of chapter 3 where these pathways are described. They may also be integrated with other biological processes described in the text.

Lines 26-28: Please check the sentences for missing letters.

Line 39: bracket missing after references

Line 82: “and all the spinal cord ; strong expression…” space not needed, please correct.

Line 119: “Taibud stages”

Lines 162-163 and line 165: some references are not formatted correctly

Line 292: please explain the acronym for PNS

Line 293: please explain the acronym for SVZ

Line 319: double repetition of the word “the”. Please correct.

Lines 364-365: check the sentence and relative brackets.

Line 369: check the sentence “…and 1 gene was found up-regulated;”

Lines 382-383: check the sentence

Line 487: please explain the acronym for NCCs the first time it appears (it is specified in line 490 instead)

Abbreviations chapter: please report all the acronyms used in the text in alphabetical order.

Author Response

We thank Reviewer 1 for the nice comments on the manuscript.

We have addressed his points as follows:

We have introduced two new Figures (Figure 1 and Figure 2). As required by the reviewer, Figure 1 shows a scheme of the functional organization of HMGA1 and HMGA2 proteins, with the sites and residues subject to possible post-translational modifications. The functional connections of these modifications to most of the developmental functions described in Chapter 3 are limited, with the possible exception of some correlation for cell cycle and apoptosis. In fact, possible studies of the relevance of these modifications in a developmental context would be important, as suggested in a sentence in the Conclusions (line 1086-1090) Figure 2 (new, suggested by Reviewer 2) shows a synopsis of Hmga1 and Hmga2 expression in a developing vertebrate embryo. Instead of splitting the original figure, we have slightly re-arranged the previous Figure 2 (now Figure 3), by labeling its different sections with letters, to make more precise reference to each section throughout the different parts of the manuscript, especially through Chapter 3, as asked by the Reviewer. We also added the names of a few genes that were not mentioned in the previous version of the figure. Figure 1 of the previous version is now Figure 4

For the minor points, we thank the Reviewer for indicating them. We have addressed all of them. Where necessary, the sentences were re-phrased in order to make the meaning clearer. For line 119 we choose to leave “tailbud stages” as we mean here the different, early and late, tailbud stages that precede the tadpole stage.

Finally, we added a list of the acronyms used in the text, as asked by the Reviewer, with the only exception of those that by consolidated usage have become proper names of genes. Some of the acronyms in the previous version have been eliminated because they occurred only once in the text.

Minor modifications were also made to improve and simplify the text and to improve the spelling.

One reference was added to the list of references.

Reviewer 2 Report

The review 'HMGA Genes and Proteins in Development and Evolution' by Vignali and Marracci gives a very extensive overview about HMGA protein and their physiological function. I find that the review is very easy to follow and in general well written. I only have a few suggestions that the authors can decide to incorporate. I find that the section on developmental expression is a bit redundant as the tissue expression is picked up at various parts of the manuscript and it felt a bit unsatisfactory to get a simple description of expression states without any functional correlation. One compromise might be to summarise this part in a figure.

The Introduction needs some editing. Some typos I found:

line 31: I don' think 'non-histonic' exist, should be non-histone
line 48, should be intrinsically
line 50-51: ‘Many of these …(enhanceosomes)’: not clear to me, please rephrase
line 60, should be: in many aspects

Author Response

We thank the Reviewer for the nice comments on the manuscript.

We understand the issue of possible redundancy of some informations on the developmental expression of the HMGA genes between Chapter 2 and Chapter 3. However, the idea was to provide a first description of the general expression of the genes in Chapter 2 to provide the basis for a better understanding of the functional part, and then to recall the most significant aspects of their expression in Chapter 3, when dealing the functional aspects. We agree that a scheme resuming the expression pattern of Hmga1 and Hmga2 could be useful and to this purpose we introduced a new figure (Figure 2) in the revised version).

Another new Figure (Figure 1 in this version) was introduced, and the previous Figure 2 (presently Fig. 3) was modified to deal with comments from Reviewer 1.

We also addressed all the minor points raised by the Reviewer, by correcting the indicated mistakes and rephrasing line 50-51 of the previous version.

Minor modifications were also made to improve and simplify the text and to improve the spelling.

One reference was added to the list of references.